# Research on Comprehensive Multi-Infrastructure Optimization in Transportation Asset Management: The Case of Roads and Bridges

**Zhang Chen [1], Yuanlu Liang [1,\*], Yangyang Wu [2] and Lijun Sun [1]**

[1]   The Key Laboratory of Road and Traffic Engineering, Ministry of Education, 4800 Cao'an Road, Jiading District, Shanghai 201804, China

[2]   Shanghai Municipal Engineering Design Institute (Group) Co., Ltd., Shanghai 200092, China

\*   Correspondence: lyl1818@tongji.edu.cn; Tel.: +86-152-1671-9779

**Abstract:** Optimization is the core of transportation asset management, but current optimization approaches are still in the stage of single infrastructure management, which seriously hinders the development and application of transportation asset management. This paper establishes a comprehensive multi-infrastructure optimization model for transportation assets consisting of roads and bridges, which is aimed at achieving the goal of transportation asset comfort, integrity, and security, taking budget funds as constraint conditions, and applying the optimization technique of goal programming and integer programming. An interactive fuzzy linear-weighted optimum-order algorithm is presented to solve the comprehensive optimization model. Finally, the comprehensive multi-infrastructure optimization model and algorithm are verified to be effective by practical data in a case study. The results indicate that the model and algorithm can provide a satisfactory and reasonable maintenance and rehabilitation schedule for transportation asset management agencies.

**Keywords:** transportation asset management; multi-infrastructure; multi-objective optimization; fuzzy method

## 1. Introduction

Traditional transportation infrastructure management emphasizes a single infrastructure. For example, PAVER was developed for pavement management by the US Army [1]; the aim of this kind of system is to provide a desired level of pavement service at the lowest overall cost [2]. PONTIS was developed for bridge management by the American Association of State Highway and Transportation Officials (AASHTO) [3]; the goal of this kind of system is to provide the most cost-effective rehabilitations and replacements for bridges with limited funds available [4]. However, the single infrastructure management model neglects characterization of the entire transportation infrastructure network, so it can only realize the maximization of partial benefits, and cannot meet the needs of modern transportation infrastructure management. Therefore, many agencies and scholars have begun to research comprehensive multi-infrastructure optimization. They consider the overall transportation infrastructure as one kind of asset, and systematically manage multiple infrastructures for overall interest [5,6]. In 1998, the US Federal Highway Administration (FHWA) established an asset management office, and proposed that all kinds of transportation infrastructure should be managed on the basis of asset attributes. An asset management primer was compiled by the FHWA, which systematically addressed the basic content of transportation asset management, such as the concept, principles, and composition [7]. In 1999, the Organization for European Cooperation and Development Working Group released a description of asset management system engineering, stating that the

objects of asset management were all components of the road network, and through comprehensive management of limited resources it could reduce overall cost and improve the quality of management and the decision-making process. [8]. In 2002, the AASHTO issued a transportation asset management guide, which defined the task and framework of asset management and emphasized integrated management by using system theory [9].

Since the asset management primer was issued by the FHWA, transportation asset management developed rapidly as a management philosophy. However, the optimization technologies of asset management developed slowly and were unable to achieve a substantive breakthrough in practical application. Some existing studies on transportation asset management optimization take single infrastructure management as the optimization target [10–12], or optimized infrastructure management by using the linear superposition method based on single infrastructure optimization [13]. Bai et al. [14] made the trade-offs between multiple objectives in the transportation asset management field using extreme points nondominated sorting genetic algorithm II, however, this method does not incorporate the temporal dimension, which is important for actual use. Wang et al. [15] established an integer programming model and a constraint programming model to optimize the coordination of a small group of pavement and bridge maintenance projects; the assumptions in the models need to be removed for more realistic applications. Bryce et al. [16] proposed a conceptual two-step approach for cross-asset resource allocation, yet no executable mathematical expressions are given. Comprehensive multi-infrastructure optimization from the perspective of a transportation infrastructure network is still lacking.

Both the FHWA's asset management primer and the AASHTO's transportation asset management guide consider transportation asset management optimization to be at the core of an asset management framework [7,9]. That is, using various methodologies and technologies to optimize the allocation of limited manpower and material resources, creating short- or long-term maintenance and rehabilitation (M&R) decision plans, and achieving the optimal overall performance of transportation assets constitute transportation asset management optimization. As shown in Table 1 [17], it is clear that roads and bridges are the most prominent types of transport infrastructure in terms of total length or total number. So, in this paper, we take roads and bridges as the research objects and discuss comprehensive multi-infrastructure optimization in transportation asset management.

**Table 1.** The 2018 China transportation assets statistics.

| Type | Railway | Road | Bridge | Inland Waterway | Port | Airport |
|---|---|---|---|---|---|---|
| **Total Length** | 131 thousand km | 4846.5 thousand km | 55.6859 thousand km | 127.1 thousand km | | |
| **Total Number** | | | 851.5 thousand | | 23,919 | 235 |

## 2. Methods

### 2.1. Selection of Optimization Objectives

In order to realize comprehensive multi-infrastructure optimization, we first need to resolve how to conduct a comprehensive evaluation for roads and bridges together; i.e., the optimization objectives should be clear. Generally, there are 3 kinds of indicators of transportation asset performance: Functional, conditional, and structural indicators [18–20]. Among them, functional indicators mainly reflect the service level, conditional indicators mainly reflect the degree of damage, and structural indicators mainly reflect the security performance of transportation assets. For users, excellent service should be reflected in the comfort and safety of transportation assets. For managers, ensuring the service life and long-term good working conditions is a strategic goal [21,22]. Combining the requirements of users and managers, this paper considers that the optimization objectives mainly include the following: While they are in service, transportation assets should provide enough comfort and safety. In terms of maintenance, there should be as little damage as possible and integrity should be ensured at the same time. Therefore, ride comfort, integrity, and security are seen as the objectives of comprehensive optimization of transportation asset management.

### 2.1.1. Transportation Asset Ride Comfort

Ride comfort is mainly related to pavement roughness. According to AASHTO road tests, about 95% of pavement service quality comes from the roughness of the road surface [23]. Many institutions and scholars have studied the relationship between comfort and the roughness of pavement; this paper establishes a model of this relationship based on a model presented by Zhou et al. [24]. The formula for calculating the transportation asset ride comfort index (TARCI) is shown in Equation (1):

$$\text{TARCI} = f(\text{IRI}) = 100e^{-0.2634\text{IRI}}, \tag{1}$$

where IRI is the international roughness index.

### 2.1.2. Transportation Asset Integrity

Integrity refers to the degree of intactness of transportation assets under conditions of maintaining normal performance levels over the designed service life, and it mainly has 2 components: The degree of intactness of pavement and structure. In single infrastructure management, the pavement condition index (PCI) is used to evaluate pavement conditions, and the bridge condition index (BCI) is used for bridge conditions [25,26]. In this paper, integrity is evaluated by the transportation asset condition index (TACI), which is calculated by using a linear-weighted method of PCI and BCI. The computational formula for TACI is shown in Equation (2):

$$\text{TACI} = \alpha \cdot \left[ \sum_{i=1}^{n-m} w_i(\text{PCI})_i + \sum_{j=n-m+1}^{n} w_j(\text{BPCI})_j \right] + (1-\alpha) \cdot \left[ \frac{1}{m} \sum_{k=1}^{m} (\text{BCI}_{sx})_k \right], \tag{2}$$

where m is the number of total bridge units, n is the number of total transportation asset units, $\alpha$ is the weight of the pavement condition for the transportation asset condition, $w_i$ is the ratio of the area of the *i*th road pavement to the whole pavement, $w_j$ is the ratio of the area of the *j*th bridge deck (including road and bridge) to the whole pavement, $(\text{PCI})_i$ is the score of the *i*th road pavement condition, $(\text{BPCI})_j$ is the score of the *j*th bridge deck condition, and $(\text{BCI}_{sx})_k$ is the score of the *k*th bridge structure condition.

### 2.1.3. Transportation Asset Security

In this paper, security is considered from the perspective of bridge structural security and evaluated by the transportation asset security index (TASI). In the field of civil engineering, structural security is generally evaluated by structural reliability [27,28]. In order to make full use of the evaluated index of single infrastructure management, the BCI is used as a bridge structural reliability index, the change of BCI is reflected by the expectation of bridge deterioration equation, and the probability distribution of bridge structural reliability is established in this paper. Transportation asset security can be evaluated by the mean of bridge structural reliability, or TASI. The computational formula for TASI is shown in Equation (3):

$$\text{TASI} = \frac{100}{m} \sum_{i=1}^{m} P_i(\text{BCI} > \text{BCI}_0) = \frac{100}{m} \sum_{i=1}^{m} \left[ 1 - \int_{-\infty}^{\text{BCI}_0} f_i(y_t) \right], \tag{3}$$

where m is the number of total bridge units, BCI is the bridge condition index, $\text{BCI}_0$ is the minimal acceptable level of BCI, $P_i(\text{BCI} > \text{BCI}_0)$ indicates the reliability of the *i*th bridge, and $f_i(y_t)$ is the probability function of the *i*th bridge in year $y_t$.

### 2.2. Establishing the Optimization Model

In pavement or bridge management, there are mainly 2 types of optimization method, mathematical programming and heuristic algorithms. Mathematical programming are exact approaches and search for optimal solutions. Wu et al. [29] established a goal programming model concerning

two conflicting objectives in pavement preservative maintenance scheduling. Moazami et al. [30] built a pavement rehabilitation and maintenance prioritization model using analytical hierarchy process and fuzzy logic. Fwa et al. [31] developed an integer programming model for network-level routine maintenance scheduling. For large-scale problems, the optimal solutions cannot be found within limited time, whereas sub-optimal or satisfactory solutions could be found when using heuristic algorithms. Morcous et al. [32] proposed an approach to determine the optimal set of maintenance alternatives using genetic algorithm. Peng et al. [33] established a bilevel program model concerning pavement maintenance fund allocation and project prioritization, and solved this model using dynamic programming and genetic algorithm. Based on the optimization method for single infrastructure management, this paper looks at transportation asset comfort, integrity, and security, considers budget funds as constraint conditions, and applies the optimization technique of goal and integer programming, based on which comprehensive multi-infrastructure optimization model is established for transportation asset management. The form of the optimization model is shown in Equations (4)–(10).

Objectives:

$$\max z_1 = \sum_{t=1}^{T}\sum_{i=1}^{N}\sum_{j=1}^{m} X_{ijt} * B_{ijt}^1, \tag{4}$$

$$\max z_2 = \sum_{t=1}^{T}\sum_{i=1}^{N}\sum_{j=1}^{m} X_{ijt} * B_{ijt}^2, \tag{5}$$

$$\max z_3 = \sum_{t=1}^{T}\sum_{i=1}^{N}\sum_{j=1}^{m} X_{ijt} * B_{ijt}^3, \tag{6}$$

subject to

$$\sum_{t=1}^{T}\sum_{i=1}^{N}\sum_{j=1}^{m} X_{ijt} * C_{ijt} \leq A, \tag{7}$$

$$\sum_{j=1}^{m} X_{ijt} = 1 \quad (i = 1, 2, \cdots N; t = 1, 2 \cdots, T), \tag{8}$$

$$X_{ijt} = \begin{cases} 1 & \text{if treatment } j \text{ is applied in segment } i \text{ in year } t \\ 0 & \text{otherwise} \end{cases} \tag{9}$$

$$B_{ijt}^1, \ B_{ijt}^2, \ B_{ijt}^3, \ C_{ijt} > 0, \tag{10}$$

where $B_{ijt}^1$, $B_{ijt}^2$, and $B_{ijt}^3$ are, respectively, the benefits of improving transportation asset ride comfort, integrity, and security by implementing treatment j in unit i in year t; $C_{ijt}$ is the cost of implementing treatment j in unit i in year t; A is the total budget in the planning period; T is the length of the planning period, normally 5 or 10 years; N is the number of total transportation asset units; and m is the total number of treatments for each unit.

*2.3. Solving the Comprehensive Optimization Model*

We used MATLAB (MathWorks, Massachusetts, USA) for modeling and solving the model. In order to represent the dominant position of the decision-maker in the maintenance and rehabilitation scheduling process, an interactive fuzzy linear-weighted optimum-order algorithm is presented to solve the comprehensive optimization model in this section. This algorithm is divided into 3 stages. First is the interactive stage of the multi-objective decision. The decision-maker selects tools and sets parameters for the trade-off analyses according to their preferences and determine the initial weight, and then provides a range of feasible solutions. Second is the stage of assigning weights

for multi-objectives. At this stage, the weights of all objectives are fuzzily assigned based on the decision-maker's preference within the given range [34]. Through the process of this stage, multiple feasible solutions in accordance with the decision-maker's preferences are acquired. Finally, the optimal solution of the comprehensive optimization model can be achieved by sorting all feasible solutions under some prescreened principles. The workflow of this algorithm is shown in Figure 1.

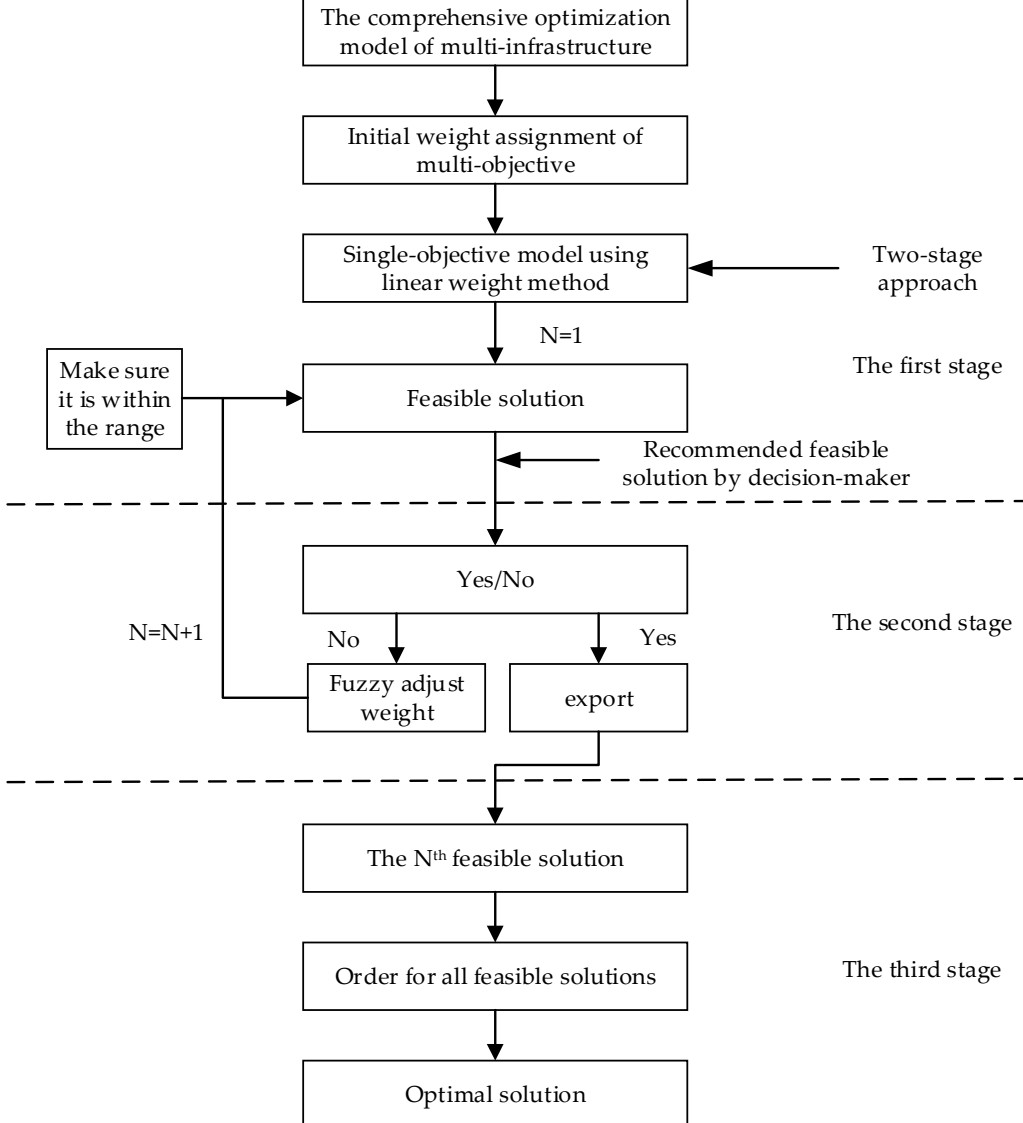

**Figure 1.** Workflow of the interactive fuzzy linear-weighted optimum-order algorithm.

### 2.3.1. Comprehensive Linear-Weighted Transportation Asset Single-Objective Optimization Model

The first stage is expressed by Equations (11)–(16):

$$\max Z = \omega_1 \sum_{t=1}^{T} \sum_{i=1}^{N} \sum_{j=1}^{m} X_{ijt} * B_{ijt}^1 + \omega_2 \sum_{t=1}^{T} \sum_{i=1}^{N} \sum_{j=1}^{m} X_{ijt} * B_{ijt}^2 + \omega_3 \sum_{t=1}^{T} \sum_{i=1}^{N} \sum_{j=1}^{m} X_{ijt} * B_{ijt}^3, \quad (11)$$

Subject to:

$$\sum_{t=1}^{T} \sum_{i=1}^{N} \sum_{j=1}^{m} X_{ijt} * C_{ijt} \leq A, \quad (12)$$

$$\sum_{j=1}^{m} X_{ijt} = 1 \quad (i = 1, 2, \cdots N; \, t = 1, 2 \cdots, T), \tag{13}$$

$$X_{ijt} = \begin{cases} 1 \text{ if treatment } j \text{ is applied in segment } i \text{ in year } t \\ 0 \text{ otherwise} \end{cases} \tag{14}$$

$$(\omega_1 \pm \Delta\omega_1) + (\omega_2 \pm \Delta\omega_2) + (\omega_3 \pm \Delta\omega_3) = 1, \; \omega_1 \in (0, \, 1), \; \Delta\omega_i \in (0, \, \alpha_i), \tag{15}$$

$$B_{ijt}^1, \; B_{ijt}^2, \; B_{ijt}^3, \; C_{ijt}, \; \alpha_i > 0 \tag{16}$$

where Z is total maintenance benefit; $\omega_1$, $\omega_2$, and $\omega_3$ are, respectively, the weight of transportation asset ride comfort, integrity, and security; and $\Delta\omega_1$, $\Delta\omega_2$, and $\Delta\omega_3$ are, respectively, the change of weight of comfort, integrity, and security, which reflects the fuzziness when the decision-maker assigns the weights.

The model above is a large integer programming problem in which the set of feasible solutions is huge. Therefore, it is necessary to simplify the model for convenient calculation and to meet practical needs. So, Equation (11) is transformed into Equation (17):

$$\begin{aligned} Z = {}& \left( \omega_1 \sum_{i=1}^{N} \sum_{j=1}^{m} X_{ij1} * B_{ij1}^1 + \omega_2 \sum_{i=1}^{N} \sum_{j=1}^{m} X_{ij1} * B_{ij1}^2 + \omega_3 \sum_{i=1}^{N} \sum_{j=1}^{m} X_{ij1} * B_{ij1}^3 \right) \\ & + \left( \omega_1 \sum_{i=1}^{N} \sum_{j=1}^{m} X_{ij2} * B_{ij2}^1 + \omega_2 \sum_{i=1}^{N} \sum_{j=1}^{m} X_{ij2} * B_{ij2}^2 + \omega_3 \sum_{i=1}^{N} \sum_{j=1}^{m} X_{ij2} * B_{ij2}^3 \right) + \cdots \\ & + \left( \omega_1 \sum_{i=1}^{N} \sum_{j=1}^{m} X_{ijT} * B_{ijT}^1 + \omega_2 \sum_{i=1}^{N} \sum_{j=1}^{m} X_{ijT} * B_{ijT}^2 + \omega_3 \sum_{i=1}^{N} \sum_{j=1}^{m} X_{ijT} * B_{ijT}^3 \right). \end{aligned} \tag{17}$$

Let $F_t(y_t)$, as shown in Equation (18), be the maintenance benefit of year t in the condition of budget $y_t$, then the total maintenance benefit Z in planning years should equal the sum of maintenance benefits of each year, as shown in Equation (19):

$$F_t(y_t) = \left( \omega_1 \sum_{i=1}^{N} \sum_{j=1}^{m} X_{ij1} * B_{ij1}^1 + \omega_2 \sum_{i=1}^{N} \sum_{j=1}^{m} X_{ij1} * B_{ij1}^2 + \omega_3 \sum_{i=1}^{N} \sum_{j=1}^{m} X_{ij1} * B_{ij1}^3 \right), \tag{18}$$

$$Z = F_1(y_1) + F_2(y_2) + \cdots F_T(y_T) = \sum_{t=1}^{T} F_t(y_t), \tag{19}$$

where $y_t$ is the budget of year t and $F_t(y_t)$ is the maintenance benefit of year t.

2.3.2. Solving the Comprehensive Weighted Single-Objective Optimization Model

A two-stage approach [35] is used to solve the comprehensive weighted single-objective optimization model in this paper. It can provide a satisfactory and reasonable maintenance schedule for the model. This approach can be divided into 2 stages, budget allocation and project distribution. Budget allocation, under certain funding, uses a genetic algorithm to find a reasonable budget allocation for each year in the planning period in order to maximize the benefit of budget. Project distribution is mainly used to arrange project implementation for each year under a given optimization result of budget allocation, solved by dynamic programming. The results of the budget allocation and project distribution models are interactional. Through iterations of the 2 models, the optimal strategy is obtained for the planning years. The process of the 2-stage approach is shown in Figure 2.

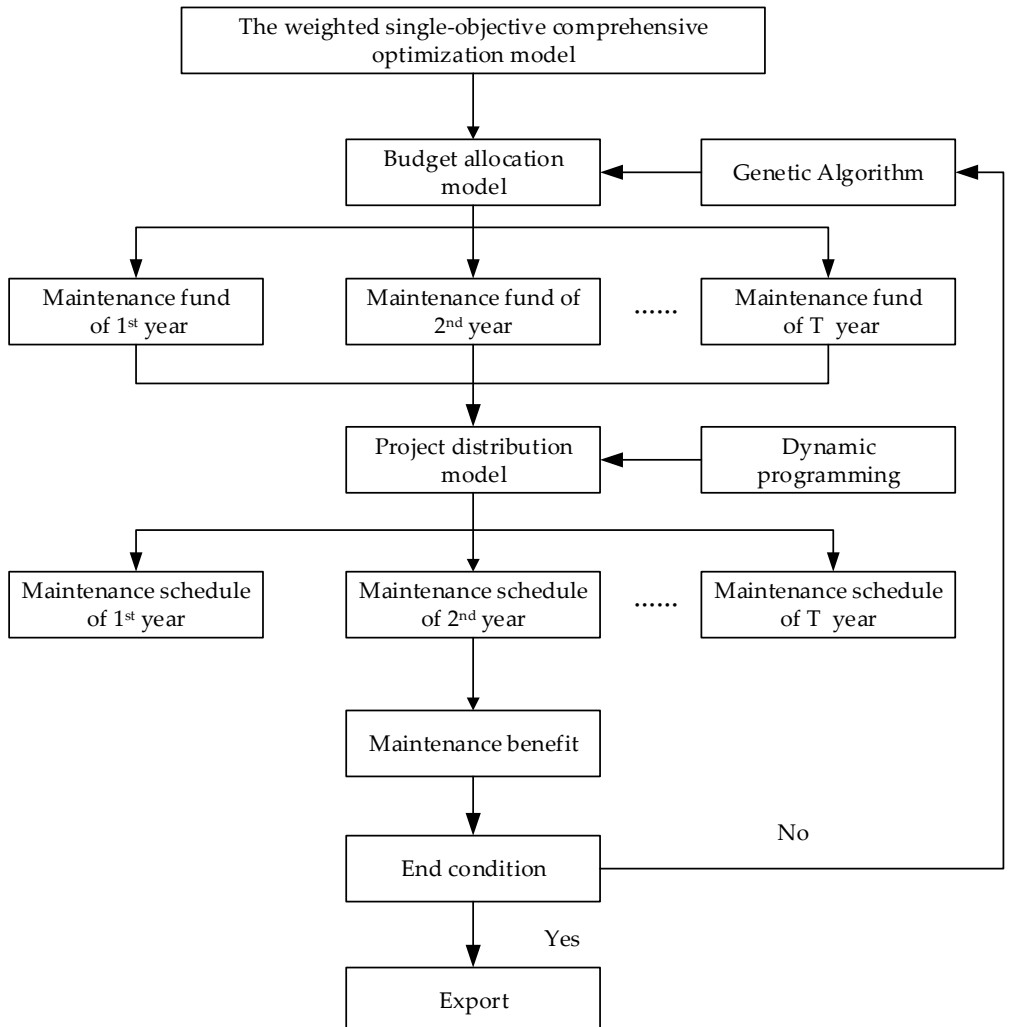

**Figure 2.** Two-stage approach for single-objective model.

### 2.3.3. Select the Optimal Solution

The optimal solution is selected by applying the optimum order method to the obtained a number of feasible solutions [36]. The total optimum number of the *i*th feasible solution $K_i$ is calculated by Equation (20) and Equation (21), the optimal solution is the *i*th solution where $K_i$ is the maximum.

$$a_{ijk} = \begin{cases} 1, & f_k(X_i) < f_k(X_j) \\ 0.5, & f_k(X_i) = f_k(X_j) \\ 0, & f_k(X_i) > f_k(X_j) \quad \text{or} \quad i = j \end{cases}, \tag{20}$$

$$K_i = \sum_{j=1}^{N} \sum_{k=1}^{P} a_{ijk}, \tag{21}$$

where $a_{ijk}$ is the optimum number when compare the *i*th solution with the *j*th solution for the *k*th objective; $f_k()$ is the *k*th objective function; $X_i$ and $X_j$ are any feasible solutions, respectively; P is the number of objective functions, N is the number of feasible solutions.

## 3. Method Verification

The methods were tested and verified through a case study with data collected from a pavement and bridge management system in Shanghai in 2008. The total number of pavement management units is 898, the total road length is 249.6 km, and the total area is 3,205,874 m². The total number of bridge management units is 399, and the total area is 241,441 m². The planning period lasts 10 years.

### 3.1. Initialization Parameters

The ranges of weights are as follows: TARCI: 0.1–0.4; TACI: 0.2–0.5; TASI: 0.2–0.5; step of assigning weight: 0.1. The conditions of feasible solutions are TARCI ≥ 20, TACI ≥ 85, TASI ≥ 99. The parameters of funds are as follows: Total budget: 600 million RMB; minimum investment for each year: 50 million RMB; maximum investment for each year: 70 million RMB; and step change of investment for each year: 1 million RMB. The genetic algorithm parameters are as follows: Reproduction rate: 0.7; mutation rate: 0.05; and crossover rate: 0.3.

### 3.2. Results

The interactive fuzzy linear-weighted optimum-order algorithm (or the new approach) was programmed with Microsoft Visual Basic 6.0. The calculation results are listed in Table 2, and the weights of TARCI, TACI, and TASI of the optimal solution are $\omega_{TARCI} = 0.3$, $\omega_{TACI} = 0.5$, and $\omega_{TASI} = 0.2$, respectively.

**Table 2.** Results of the new approach. TARCI, transportation asset ride comfort index; TACI, transportation asset condition index; TASI, transportation asset security index.

| Year | Number of Segments Treated | Total Maintenance Cost (Million RMB) | Transportation Asset Performance | | |
|---|---|---|---|---|---|
| | | | TARCI | TACI | TASI |
| 2008 | 201 | 68 | 25.28 | 90.55 | 99.37 |
| 2009 | 105 | 53.99 | 24.57 | 89.85 | 99.7 |
| 2010 | 112 | 69.01 | 25.81 | 89.36 | 99.94 |
| 2011 | 148 | 69.95 | 29.46 | 89.65 | 99.93 |
| 2012 | 131 | 60 | 30.76 | 89.95 | 99.9 |
| 2013 | 129 | 49.9 | 32.18 | 90.04 | 99.84 |
| 2014 | 132 | 53.92 | 33.47 | 90.54 | 99.97 |
| 2015 | 108 | 48.77 | 33.2 | 90.47 | 99.66 |
| 2016 | 86 | 33.37 | 32.32 | 90.01 | 99.55 |
| 2017 | 61 | 29.13 | 30.46 | 89.07 | 99.4 |
| Total | 1213 | 536.04 | 29.75 | 89.95 | 99.73 |

The new approach in this paper is compared to the current pavement and bridge maintenance approach (or the conventional approach), in which maintenance would be conducted once the PCI or BCI is lower than 75, currently adopted by pavement and bridge management agencies in Shanghai. The calculated results of the conventional approach are listed in Table 3. For comparison purposes, the calculated results of the new approach and the conventional approach in Tables 2 and 3 are shown in Figures 3 and 4.

**Table 3.** Results of the conventional approach.

| Year | Number of Segments Treated | Total Maintenance Cost (Million RMB) | Transportation Asset Performance | | |
| --- | --- | --- | --- | --- | --- |
| | | | TARCI | TACI | TASI |
| 2008 | 308 | 274.84 | 25.32 | 93.11 | 99.99 |
| 2009 | 45 | 33.73 | 24.55 | 91.87 | 99.99 |
| 2010 | 86 | 47.36 | 25.8 | 91.19 | 99.99 |
| 2011 | 133 | 49.81 | 29.46 | 91.32 | 99.99 |
| 2012 | 125 | 50.4 | 30.76 | 91.6 | 99.98 |
| 2013 | 130 | 51.69 | 32.18 | 91.73 | 99.98 |
| 2014 | 132 | 59.08 | 33.46 | 92.25 | 99.98 |
| 2015 | 102 | 38.26 | 33.19 | 91.99 | 99.96 |
| 2016 | 92 | 36.25 | 32.32 | 91.65 | 99.95 |
| 2017 | 87 | 60.08 | 30.46 | 91.1 | 99.95 |
| Total | 1240 | 701.5 | 29.75 | 91.78 | 99.98 |

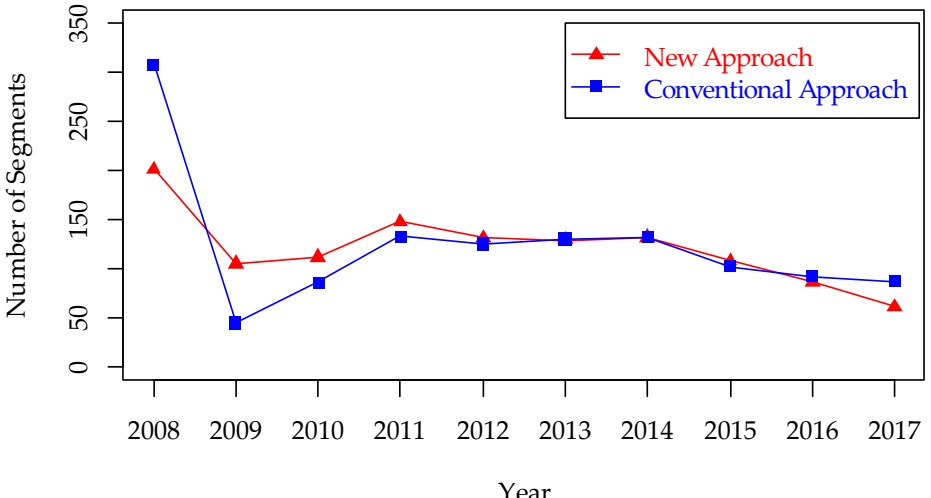

**Figure 3.** Number of segments treated each year.

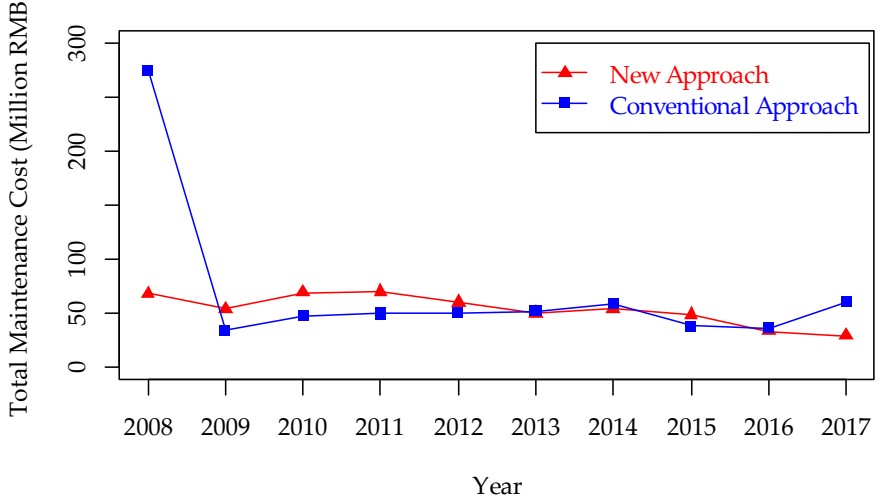

**Figure 4.** Total maintenance cost for each year.

## 4. Discussion

Based on the case study results, the following observations can be made. The total maintenance cost in the planning period was significantly reduced after adopting the new approach (from ¥701.5

million RMB to ¥536.04 million RMB, a 24% decrease). In terms of maintenance benefits, though the transportation asset performance indices in the planning period decreased after adopting the new approach (average TARCI was the same, the average TACI dropped 2%, and the average TASI dropped 0.3%), the total maintenance cost in the planning period dropped 24%, so the optimization results could be accepted by decision-makers. In terms of the practicability of the optimization results, with the conventional approach, the budget demand and number of segments treated in each planning year are significantly different (maximal budget demand is ¥274.84 million RMB and maximal number of segments to be treated is 308 in the first year, while minimal budget demand is ¥33.73 million RMB and minimal number of segments to be treated is 45 in the second year). With the new approach, the budget demand and number of segments treated in each planning year are well controlled, and the difference in each year's budget and maintenance segments effectively decreases (maximal budget demand is ¥69.95 million RMB in the fourth year and maximal number of segments treated is 201 in the first year, while minimal budget demand is ¥29.13 million RMB and minimal number of segments treated is 61 in the 10th year). This is a good fit for the actual financial plans of transportation asset management agencies.

No general principles are devised to incorporate new performance indicators into TARCI, TACI, or TASI, so when other types of transportation assets need to be considered, expert knowledge is needed for the redesign of the objectives, which negatively affects the expansibility of the proposed model and algorithm.

## 5. Conclusions

This paper takes roads and bridges as an example and established a comprehensive multi-infrastructure optimization model, which has the goals of transportation asset ride comfort, integrity, and security, taking budget funds as constraint conditions. This model can be solved by firstly assigning fuzzy weights to all objectives, then applying a two-stage approach to solve the comprehensive weighted single-objective optimization model, and finally using an optimum-order method to select the optimal solution. A case study shows the effectiveness of the optimization model and algorithm: The total maintenance cost is reduced sharply with only minor and acceptable decreases in the infrastructure performances. The maintenance and rehabilitation schedule is more balanced and, thus, more applicable for actual use. Consequently, the model and algorithm proposed in this paper can serve as an efficient tool for transportation asset management agencies, though further modeling techniques are needed for extended use.

**Author Contributions:** Conceptualization, C.-Z. and L.-Y.L.; methodology, C.-Z. and W.-Y.Y.; software, C.-Z.; validation, L.-Y.L. and S.-L.J.; formal analysis, C.-Z. and L.-Y.L.; investigation, W.-Y.Y. and S.-L.J.; resources, C.-Z.; data curation, C.-Z. and W.-Y.Y.; writing—original draft preparation, C.-Z., L.-Y.L. and W.-Y.Y.; writing—review and editing, L.-Y.L. and S.-L.J.; visualization, L.-Y.L.; supervision, S.-L.J.; project administration, C.-Z.; funding acquisition, C.-Z.

**Funding:** This research was funded by the National Natural Science Foundation of China (#71471134).

**Acknowledgments:** Thanks to the Road Administration of Shanghai for kindly providing the historical datasets for the case study.

**Conflicts of Interest:** The authors declare no conflict of interest. The funders had no role in the design of the study; in the collection, analysis, or interpretation of data; in the writing of the manuscript; or in the decision to publish the results.

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
