# Peer review of "Research on Comprehensive Multi-Infrastructure Optimization in Transportation Asset Management: The Case of Roads and Bridges"

_sustainability, doi:10.3390/su11164430_

Round 1

Reviewer 1 Report

The article is written according to requirements.

The introduction include not all relevant references. The methods were tested in 2008 but the authors have submitted only 2 articles to the bibliography [20, 2007] and [31, 2014].

The conclusions can be refined and expanded.

The article does not describe what software is used for modeling. It is necessary to write.

Author Response

Q1: The introduction include not all relevant references. The methods were tested in 2008 but the authors have submitted only 2 articles to the bibliography [20, 2007] and [31, 2014].

A1: We have introduced more relevant references to address the optimization methods both in the introduction part and methods part, these are reference [14-16] and [33].

Q2: The conclusions can be refined and expanded.

A2: We have simplified the original expressions and add more content drawn from the discussion part to it.

Q3: The article does not describe what software is used for modeling. It is necessary to write.

A3: We used MATLAB (MathWorks, Massachusetts, USA) for modeling. Now we write it into the article.

Reviewer 2 Report

Page 2, lines 61, 62: “…..we take roads and bridges as the research objects and discuss comprehensive multi infrastructure optimization in transportation asset management…”. It will be for the benefit of the reader if you can provide a list of all the transportation infrastructure assets and also if you can provide a more detailed justification (than the one provided in lines 62, 63 and 64) about the reason(s) for which you select only “roads” and “bridges” for your research.

The title of the paper should change to “Research on Comprehensive Multi-Infrastructure  Optimization in Transportation Asset Management: The case of Roads and Bridges” in order to better reflect the content of the paper.

Page 4, lines 137, 138: “…..At this stage, the decision-maker sets the initial weight for multi-objectives according to their experience,…..” Can you please provide some more details about the “initial weight” ? Are weights subjective ? To what extent ?

Conclusions: As far as the “model and algorithm proposed” is concerned, please refer to their limitations and constraints. How these affect the general applicability of your model and algorithm ?

Minor comments:

Page 1, line 30: Please add some more detailed text concerning the 4 references [1-4].

Page 2, line 88: Heading “2.1.2. Transportation Asset Integrity” must be on page 3.

Page 3, lines 97, 98, 99, 100 and 101: Line spacing is different when it is compared to the rest of the text.

Page 3, line 119: Please add some more detailed text concerning the 5 references [25-29].

Page 4, lines 127, 128 and 129: Line spacing is different when it is compared to the rest of the text.

Reference [32] does not appear within the text although it appears in the references list at the end of the paper.

Page 7, line 186: Heading “3.2. Results” must be on page 8.

Figure 3, Figure 4: The fonts used in labels, in X-axis and in Y-axis are different when they are compared to the rest of the text.

Page 8, Table 1: Total of column 2 is 1213 and not 1231.    

Page 8, Table 1: Total of column 3 is 701.5 and not 699.01

Author Response

Q1: Page 2, lines 61, 62: “…..we take roads and bridges as the research objects and discuss comprehensive multi infrastructure optimization in transportation asset management…”. It will be for the benefit of the reader if you can provide a list of all the transportation infrastructure assets and also if you can provide a more detailed justification (than the one provided in lines 62, 63 and 64) about the reason(s) for which you select only “roads” and “bridges” for your research.

A1: We have provided the 2018 China transportation asset statistics and changed the description in the original lines 62, 63 and 64 accordingly.

Q2: The title of the paper should change to “Research on Comprehensive Multi-Infrastructure  Optimization in Transportation Asset Management: The case of Roads and Bridges” in order to better reflect the content of the paper.

A2: We have changed the title accordingly.

Q3: Page 4, lines 137, 138: “…..At this stage, the decision-maker sets the initial weight for multi-objectives according to their experience,…..” Can you please provide some more details about the “initial weight” ? Are weights subjective ? To what extent ?

A3: The initial weight is usually determined by trade-off analyses, which is a mixture of quantitative and qualitative analysis. The subjective part is that the decision makers choose the tool and set the parameters for trade-off analyses, which reflects their preferences. We have changed the original expression and add more details for the initial weight determination.

Q4: Conclusions: As far as the “model and algorithm proposed” is concerned, please refer to their limitations and constraints. How these affect the general applicability of your model and algorithm ?

A4: The model and algorithm proposed has some difficulties in considering other types of transportation infrastructures, we have added more details in the discussion part and concluded it in the conclusion part.

Minor comments:

Q5: Page 1, line 30: Please add some more detailed text concerning the 4 references [1-4].

A5: We have added more details to it.

Q6: Page 2, line 88: Heading “2.1.2. Transportation Asset Integrity” must be on page 3.

A6: We have changed the layout accordingly.

Q7: Page 3, lines 97, 98, 99, 100 and 101: Line spacing is different when it is compared to the rest of the text.

A7: This is due to the use of Mathtype. Now we have transformed the Mathtype characters into ordinary characters to make sure that the line spacing is always consistent. This change is also made to other places where analogous situation exists.

Q8: Page 3, line 119: Please add some more detailed text concerning the 5 references [25-29].

A8: We have added more detailed text for the original reference [25-28] and removed the original reference [29] for there are already 3 references concerning the mathematical programming. We replaced the original reference [29] for another reference to exemplify the use of heuristic algorithms.

Q9: Page 4, lines 127, 128 and 129: Line spacing is different when it is compared to the rest of the text.

A9: We have transformed the Mathtype characters into ordinary characters to make sure that the line spacing is always consistent.

Q10: Reference [32] does not appear within the text although it appears in the references list at the end of the paper.

A10: We are terribly sorry that we missed the 2.3.3 section in the preparation process, now we added it to the article and the original reference [32] is visible.

Q11: Page 7, line 186: Heading “3.2. Results” must be on page 8.

A11: We have changed the layout accordingly.

Q12: Figure 3, Figure 4: The fonts used in labels, in X-axis and in Y-axis are different when they are compared to the rest of the text.

A12: We have changed the theme of the characters in Figure 3 and Figure 4, now they are in consistent with the main body of this article.

Q13: Page 8, Table 1: Total of column 2 is 1213 and not 1231.    

A13: We have corrected the calculation error.

Q14: Page 8, Table 1: Total of column 3 is 701.5 and not 699.01

A14: We have corrected the calculation error.

Round 2

Reviewer 2 Report

All areas of concern have been addressed as advised.